# High-Resolution Secretome Analysis of Chemical Hypoxia Treated Cells Identifies Putative Biomarkers of Chondrosarcoma

**DOI:** 10.3390/proteomes10030025

**Published:** 2022-07-20

**Authors:** Donatella Pia Spanò, Simone Bonelli, Matteo Calligaris, Anna Paola Carreca, Claudia Carcione, Giovanni Zito, Aldo Nicosia, Sergio Rizzo, Simone Dario Scilabra

**Affiliations:** 1Proteomics Group of Fondazione Ri.MED, Department of Research IRCCS ISMETT, via Ernesto Tricomi 5, 90145 Palermo, Italy; dspano@fondazionerimed.com (D.P.S.); sbonelli@fondazionerimed.com (S.B.); mcalligaris@fondazionerimed.com (M.C.); apcarreca@fondazionerimed.com (A.P.C.); 2STEBICEF (Dipartimento di Scienze e Tecnologie Biologiche Chimiche e Farmaceutiche), Università degli Studi di Palermo, Viale delle Scienze Ed. 16, 90128 Palermo, Italy; 3Department of Pharmacy, University of Pisa, Via Bonanno 6, 56126 Pisa, Italy; 4Fondazione Ri.MED, Department of Research IRCCS ISMETT, via Ernesto Tricomi 5, 90145 Palermo, Italy; ccarcione@fondazionerimed.com; 5Research Department, IRCSS ISMETT (Instituto Mediterraneo per i Trapianti e Terapie ad Alta Specializzazione), 90127 Palermo, Italy; giovanni.zito31@gmail.com; 6Institute for Biomedical Research and Innovation-National Research Council (IRIB-CNR), Via Ugo La Malfa 153, 90146 Palermo, Italy; aldo.nicosia@irib.cnr.it; 7Medical Oncology Unit, IRCCS ISMETT (Istituto Mediterraneo per i Trapianti e Terapie ad Alta Specializzazione), 90127 Palermo, Italy; srizzo@ismett.edu

**Keywords:** chondrosarcoma, hypoxia, proteomics, extracellular vesicles, GAPDH, biomarkers

## Abstract

Chondrosarcoma is the second most common bone tumor, accounting for 20% of all cases. Little is known about the pathology and molecular mechanisms involved in the development and in the metastatic process of chondrosarcoma. As a consequence, there are no approved therapies for this tumor and surgical resection is the only treatment currently available. Moreover, there are no available biomarkers for this type of tumor, and chondrosarcoma classification relies on operator-dependent histopathological assessment. Reliable biomarkers of chondrosarcoma are urgently needed, as well as greater understanding of the molecular mechanisms of its development for translational purposes. Hypoxia is a central feature of chondrosarcoma progression. The hypoxic tumor microenvironment of chondrosarcoma triggers a number of cellular events, culminating in increased invasiveness and migratory capability. Herein, we analyzed the effects of chemically-induced hypoxia on the secretome of SW 1353, a human chondrosarcoma cell line, using high-resolution quantitative proteomics. We found that hypoxia induced unconventional protein secretion and the release of proteins associated to exosomes. Among these proteins, which may be used to monitor chondrosarcoma development, we validated the increased secretion in response to hypoxia of glyceraldehyde 3-phosphate dehydrogenase (GAPDH), a glycolytic enzyme well-known for its different functional roles in a wide range of tumors. In conclusion, by analyzing the changes induced by hypoxia in the secretome of chondrosarcoma cells, we identified molecular mechanisms that can play a role in chondrosarcoma progression and pinpointed proteins, including GAPDH, that may be developed as potential biomarkers for the diagnosis and therapeutic management of chondrosarcoma.

## 1. Introduction

Chondrosarcoma is a group of primary bone malignancies with dissimilar clinical outcomes. It accounts for approximately 20% of malignant bone tumors, making chondrosarcoma the second most common bone tumor in adults [1,2]. Chondrosarcoma is resistant to most chemo- and radiotherapies, thus surgical resection is still considered as the major treatment for this type of tumor. Consequently, histological classification of chondrosarcoma grade is the most used tool to assess its prognosis. However, this evaluation is performed by a histopathologist, and therefore results are operator-dependent and tumor grading is not objective. Proper assessment of the tumor grade is essential, as its treatment after surgical resection can considerably vary between low- and intermediate-grade chondrosarcomas [3,4]. It is widely ascertained that more accurate methods to assess chondrosarcoma prognosis are needed, and, for this reason, there is a growing interest in finding less invasive, yet specific and accurate prognostic biomarkers to estimate chondrosarcoma grade and improve clinical and therapeutical management [4,5]. In addition, high-grade tumors often metastasize leading to death and dissecting the molecular mechanisms that render chondrosarcoma unresponsive to chemotherapy and radiotherapy, including slow growth, abundant deposition of extracellular matrix (ECM) and poor vascularity, is urgently needed to develop effective therapeutic strategies for this tumor.

Similarly to other tumors, chondrosarcoma malignancy arises when metastasis spreads due to the neoangiogenic process in support of uncontrolled growth [6]. Hypoxia, which is a condition of inadequate oxygen availability involved in many pathophysiological processes, plays a pivotal role in the evolution of cartilage tumors due to the phenotypic features of this tissue that are hypoxic and avascular. The hypoxic response is mainly ascribed to hypoxia-inducible factors (HIFs), in particular the heterodimers HIF-1α and HIF-1β. While HIF-1β is constitutively expressed, HIF-1α expression is oxygen-sensitive. In the presence of oxygen, HIF-1α is hydroxylated by prolyl hydroxylase domain-containing proteins (PHDs) and then targeted for proteasomal degradation. Conversely, in the absence of oxygen, HIF-1α moves into the nucleus, where it dimerizes with HIF-1β, and activates the expression of hypoxia-responsive genes [7,8]. Among hypoxia-responsive genes, HIF-1α directly promotes the expression of most genes involved in angiogenesis and metabolism, such as glycolytic enzymes and glycogenic enzymes, to further improve cancer proliferation and invasiveness [8,9]. Recently, several studies have shown the association of HIF-1α expression with the outcome of bone tumors. High HIF-1α expression correlates with tumor differentiation, clinical stage and metastasis. Furthermore, high HIF-1α expression is strongly associated with the microvessel density of bone tumors. In malignant chondrosarcoma, HIF-1α activates VEGF-A expression, a cytokine that plays a crucial role in neoangiogenic process [6], but also CXCR4 and MMP1, which mediate cell migration and invasion [10], as a response to the hypoxic microenvironment. Association studies between HIF-1α, its downstream targets and prognosis in patients with bone tumors have shown unfavorable overall survival in patients with a high expression of HIF-1α [11]. In conclusion, mounting evidence is emerging that HIF-1α strongly correlates with the grade and prognosis of chondrosarcoma.

In this report, we stimulated the HIF-1α-dependent hypoxic response in chondrosarcoma cells using dimethyloxalylglycine (DMOG), a chemical compound that stabilizes HIF-1α [12,13,14]. Then, we used unbiased high-resolution proteomics to identify proteins that were differentially secreted by chondrosarcoma in response to hypoxia. Our study revealed major changes in the secretome of DMOG-treated chondrosarcoma cells, which were linked to an increase in exosome and unconventional protein secretion. Given the role of hypoxia in chondrosarcoma development, changes induced by DMOG may resemble the pathological features of tumor development. In addition, we identified a number of proteins whose secretion was increased in response to hypoxia, including glyceraldehyde 3-phosphate dehydrogenase (GAPDH), which may be further developed as prognostic biomarkers of chondrosarcoma.

## 2. Materials and Methods

### 2.1. DMOG Treatment of SW 1353 Cells

SW 1353 cells [also referred to as SW-1353 or HTB94], a human chondrosarcoma cell line, were kindly provided by Prof Hideaki Nagase, and were originally purchased from American Culture Type Collection (Manassas, VA, USA). SW 1353 were cultured in Dulbecco’s Modified Eagle’s Medium (DMEM) containing 1% L-Glutamine, 1% Penicillin and Streptomycin (Pen-Strep), 1% Sodium Pyruvate and 10% Fetal Bovine Serum (FBS) at 37 °C, 5% CO_2_. SW 1353 cells were seeded in 6-well plates and grown in complete medium, then they were washed in Dulbecco’s Phosphate Buffered Saline (DPBS) and incubated with 500 μM DMOG (Sigma-Aldrich, St. Louis, MO, USA) in serum-free DMEM for 24 h. Equal volumes of dimethyl sulfoxide (DMSO, Sigma-Aldrich, St. Louis, MO, USA), the agent in which DMOG was dissolved, were applied to SW1353 cells as a negative control.

### 2.2. Sample Processing and Mass Spectrometry Analysis

Three biological replicates were performed for DMOG-treated and control cells. Conditioned media of DMOG-treated or control cells were harvested when 90% confluence was reached, then centrifuged (14,000× *g* for 10 min) to remove cell debris and concentrated by Vivaspin protein concentrator spin columns with a 10 kDa molecular weight cut-off (Sartorius, Göttingen, Germany). Proteins were subjected to *filter-aided sample preparation* (FASP) [15]. Briefly, proteins were reduced by the addition of 1 M Dithiothreitol (DTT) in 100 mM Tris/HCl, 8 M urea pH 8.5 for 30 min at 37 °C. Proteins were then alkylated in 50 mM iodoacetamide (IAA) for 5 min at room temperature and washed twice in 100 mM Tris/HCl, 8 M urea pH 8.0 at 14,000× *g* for 30 min. 10 μg of protein per sample were digested with 0,2 μg LysC (Promega, Madison, WI, USA) in 25 mM Tris/HCl, 2 M urea pH 8.0 overnight (enzyme to protein ratio 1:50) and with 0,1 μg trypsin (Promega, Madison, WI, USA) in 50 mM ammonium bicarbonate for 4 h (enzyme to protein ratio 1:100) [16]. Generated peptides were desalted by *stop-and-go extraction* (STAGE) on reverse phase C18 (Supelco Analytical Products, part of Sigma-Aldrich, Bellefonte, PA, USA), as previously described [17], and eluted in 40 μL of 60% acetonitrile in 0.1% formic acid. The volume was reduced in a SpeedVac (Thermo Fisher Scientific, Waltham, MA, USA) and the peptides were resuspended in 20 μL of 0.1% formic acid, prior to being analyzed by LC-MS/MS. 5 μL peptides were separated on an Acclaim PEPMap C18 column (50 cm × 75 µm ID, Thermo Scientific) with 250 nL/min flow using a 220 min binary gradient of water and acetonitrile (from 2% to 95% acetonitrile in water). Then, peptides were analyzed using a Dionex Ultimate 3000 RSLCnano LC system coupled online via a Nanospray Flex Ion Source (Thermo Scientific) with a Q-Exactive mass spectrometer (Thermo Fisher Scientific, Waltham, MA, USA). Peptide intensities were quantified by using label-free quantification (LFQ) using data-dependent acquisition (DDA). Full MS scans were acquired at a resolution of 70,000 (*m*/*z* range: 300–1400; automatic gain control (AGC) target: 1 × 10^6^; max injection time 50 ms). The DDA was used on the 10 most intense peptide ions per full MS scan for peptide fragmentation (resolution: 17,500; isolation width: 2 *m*/*z*; AGC target: 1 × 10^5^; normalized collision energy (NCE): 25%, max injection time 55 ms). A dynamic exclusion of 120 s was used for peptide fragmentation.

### 2.3. Proteomic Data Analysis

The data were normalized and analyzed using Maxquant software (maxquant.org, Max-Planck Institute Munich, version 2.0.1.0 [18]) and searched against a reviewed canonical FASTA database of homo sapiens, as previously described [19,20]. Trypsin was defined as protease, and two missed cleavages were allowed for the database search. The option “first search” was used to recalibrate the peptide masses within a window of 20 ppm. For the main search, peptide and peptide fragment mass tolerances were set to 4.5 and 20 ppm, respectively. Carbamidomethylation of cysteine was defined as a static modification. Protein acetylation at the N-terminus and oxidation of methionine were set as variable modifications. Only unique peptides were used for label-free quantification, and the match-between-runs option was used. The Perseus software platform (http://www.perseus-framework.org (accessed on 7 July 2022); copyright of Max Planck Institute of Biochemistry- Martinsried- Munich; Germany) was used to perform statistical analysis and evaluate changes in protein levels between DMOG-treated and control cells [21]. LFQ values were log2 transformed and a two-sided Student’s *t*-test was used for the statistical analysis of three DMOG-treated samples versus controls.

### 2.4. EVs Isolation and Characterization

For each experiment, SW 1353 cells were grown in complete medium in 175 cm^2^ flasks, washed once in DPBS and incubated with serum-free DMEM supplemented with 500 μM DMOG or DMSO as a control. Conditioned media were collected from 90% confluent cells and further processed for EVs isolation [22]. Briefly, conditioned media were sequentially centrifuged at 4 °C at 300× *g* for 10 min and 2000× *g* for 20 min, to get rid of cells and debris. Then, supernatants were centrifuged at 10,000× *g* for 40 min and at 100,000× *g* for 70 min. Pelleted EVs were collected in DPBS and analyzed by nanoparticle tracking analysis (NTA) to assess the number of vesicles per cells and the diameter, using a NanoSight NS3000 (Malvern Panalytical, part of Spectris plc, Malvern, Worcestershire, United Kingdom).

### 2.5. Validation of GAPDH Levels by Western Blotting

SW 1353 cells were grown in 6-well plates and then incubated in serum-free medium with 500 μM DMOG or equal volume of DMSO as a control. After 24 h, when the confluence reached about 90%, conditioned media were harvested, separated from EVs by sequential centrifugation, and proteins precipitated with 5% *v*/*v* trichloroacetic acid (Sigma, Aldrich, St. Louis, MO, USA) before being resuspended in a Laemmli sample buffer (Bio-Rad, Hercules, CA, USA). Cells were collected with a STET lysis buffer (50 mM Tris, pH 7,5, 150 mM NaCl, 2 mM EDTA, 1% Triton), containing protease inhibitor cocktail (1:100, P-2714, Sigma, Aldrich, St. Louis, MO, USA). Protein concentration was measured by using a colorimetric 660 nm microBCA assay (Thermo Fisher Scientific, Waltham, US). EVs isolated from DMOG-treated or control cells by sequential centrifugation were resuspended in the Laemmli sample buffer (Bio-Rad, Hercules, CA, USA). Proteins from conditioned media, lysates and EVs were loaded onto an acrylamide gel and analyzed using SDS-PAGE electrophoresis, followed by immunoblotting. The Trans-Blot Turbo system (Bio-Rad, Hercules, CA, USA) was used for protein transfer (Standard protocol: 30 min, 1.0 A, 25 V). The following antibodies were used: anti-GAPDH (88845, Cell Signaling, Danvers, MA, USA), anti-calnexin (ADI-SPA-860-F, ENZO lifescience, Farmingdale, NY, USA). A goat anti-Rabbit HRP-conjugated secondary antibody from Promega was used.

### 2.6. Assessment of SW 1353 Cell Viability upon DMOG Treatment

SW 1353 cells were grown in 96-well plates and then incubated for 24 h in serum-free medium supplemented with 500 μM DMOG, 1 µM doxorubicin or DMSO as a negative control. Then, cell viability was assessed using a Cell Titer Glo 2.0 kit (Promega, Fitchburg, WI, USA), which determines the number of viable cells in a culture by quantifying intracellular ATP, according to the manufacturer’s instruction. Finally, cells were grown until 90% confluence in 6-well plates and were then treated with 500 μM DMOG, 1 µM doxorubicin or DMSO for 24 h in serum-free DMEM. Secretion of GAPDH in these cells was evaluated by western blotting as previously described.

## 3. Results

### 3.1. Quantitative Proteomics Identified Differences in the Secretome of Hypoxia-Induced Chondrosarcoma Cells

In order to identify the effects induced by hypoxia on proteins secreted by chondrosarcoma SW 1353 cells, we treated the cells with DMOG for 24 h, before applying the conditioned media to a high-resolution mass spectrometry-based workflow. By using this workflow, which comprises tryptic digestion by *filter-aided sample preparation* (FASP, [15]) and protein analysis via LC-MS/MS followed by label-free quantification, we detected 856 proteins in the secretome of both DMOG-treated and control SW 1353 cells (Figure 1A, Appendix A).

Proteins were arbitrarily considered as significantly altered upon DMOG treatment in the secretome of SW 1353 cells when their fold change was at least 50% (displayed as the two vertical dashed lines in Figure 1A), and the *p*-value of their change was below 0.05 (displayed as the horizontal dashed line in Figure 1A). This yielded 103 altered proteins, of which 77 proteins were more abundant and 26 were less abundant in the secretome of DMOG-treated cells (Figure 1A, Table 1 and Table 2). Among these 103 altered proteins, 61 were cytoplasmic proteins, 36 were nuclear proteins, 19 were secreted proteins and 8 were cell membrane proteins (according to Uniprot annotation, Figure 1B, Appendix A). A further computational analysis using the TRANSFAC database on transcription factors [23,24] confirmed that such secretome alterations were majorly induced by the activation of HIF-1α (Table 3).

### 3.2. Enrichment Analysis Showed Hypoxia Enriched EVs and Exosomes Proteins Compartmentalization

DMOG altered levels of a heterogenous group of proteins in the secretome of chondrosarcoma cells, thus, in order to identify molecular pathways or biological processes regulated by hypoxia in these cells, altered proteins were analyzed using the Enrichr web server for gene ontology (GO) [25,26,27]. The GO-Biological Processes analysis found that the major group of proteins regulated by hypoxia were those associated with extracellular matrix (ECM) organization (GO:0030198—Table 4). The GO-Cellular Components analysis displayed the secretory granule lumen as the most regulated group of proteins (GO:0034774—Table 5). In line with this, a further computational analysis by using the Jensen COMPARTMENTS database of protein subcellular localization identified as associated to extracellular exosomes and extracellular vesicles the large majority of proteins altered by DMOG treatment (Table 6) [28].

In conclusion, our proteomic analysis identified major changes in the secretome of SW 1353 chondrosarcoma cells in response to chemically-induced hypoxia. A GO enrichment analysis revealed that hypoxia mostly affected unconventionally secreted and exosome-associated proteins, and proteins involved in the organization of the ECM.

### 3.3. Hypoxia Increased Secretion of Extracellular Vesicles

Unbiased proteomics found a large number of exosome-associated proteins being altered in the secretome of chondrosarcoma SW 1353 cells upon DMOG treatment. Thus, we analyzed whether hypoxia could affect the release of exosomes. Extracellular vesicles (EVs) were isolated from the conditioned media of DMOG-treated or control cells by differential ultracentrifugation and characterized by nanoparticle tracking analysis (NTA). We observed an overall size distribution of EVs ranging from 110 nm to 226 nm (Figure 2A). The NTA profile of EVs from SW 1353 cells showed a major peak of about 139 nm, indicating that the most abundant group of extracellular vesicles in these cells are exosomes (which are typically < 150 nm in diameter, [29]). Similarly, the most abundant group of EVs in DMOG-treated cells are exosomes, with a peak of 144 nm in diameter (Figure 2A). Furthermore, DMOG treatment induced no significative variation in the modal distribution of EVs isolated from conditioned media of SW 1353 cells (Figure 2B). Conversely, the concentration of EVs isolated from conditioned media of DMOG-treated SW 1353 cells was about 2.4 times higher than the concentration of EVs isolated from control cells (Figure 2C), suggesting that hypoxia increased the secretion of exosome-associated proteins by stimulating the release of exosomes.

### 3.4. GAPDH: Validation of Secretome Analysis and Enrichment Analysis

Glyceraldehyde 3-phosphate dehydrogenase (GAPDH) is a key enzyme in glycolysis that catalyzes the first step of the pathway by converting D-glyceraldehyde 3-phosphate (G3P) into 3-phospho-D-glyceroyl phosphate [30]. GAPDH expression and protein function is altered in many cancer cell types, mainly in response to hypoxic stress in the tumor microenvironment [9,31,32], and is related to tumor progression and invasiveness [33]. GAPDH is mainly cytosolic, but it can also be unconventionally secreted [34,35] and found in biological fluids [36]. Other than being released through exosomes [37], GAPDH was reported to play a role in their assembly and secretion [38]. Secretome analysis found that extracellular levels of GAPDH were increased upon DMOG treatment (Figure 1A, Table 1). Thus, we validated the increase of GAPDH as an example of exosome-associated and unconventionally secreted protein in response to hypoxia, for which reliable antibodies and ELISAs are currently available. In line with proteomics results, extracellular levels of GAPDH increased upon treatment of SW 1353 cells with DMOG (Figure 3).

Conditioned media from DMOG-treated and control cells underwent serial centrifugations to isolate the extracellular vesicles. Levels of GAPDH in the EVs-depleted conditioned media and EV fraction were analyzed by western blotting (Figure 3A). DMOG treatment increased GAPDH levels in both fractions, suggesting that hypoxia promotes unconventional secretion of the protein through exosomes and through an exosome-independent secretion pathway.

Next, we investigated whether DMOG could stimulate secretion of GAPDH by a mechanism related to cell stress or apoptosis. Thus, we first assessed the viability of DMOG-treated SW 1353 cells. As shown in Figure 3B, DMOG had a minimal, yet significant effect on cell viability. Doxorubicin, a widely used chemotherapeutic agent known to induce apoptosis on cultured cells [39,40], decreased the viability of SW 1353 cells by over 30% under similar conditions (Figure 3B). Nevertheless, in contrast to DMOG, doxorubicin treatment did not increase levels of GAPDH in the conditioned media of SW1353, indicating that its secretion in response to DMOG treatment was induced by hypoxia, rather than by an unrelated stress-dependent mechanism (Figure 3C).

## 4. Discussion

Chondrosarcoma is the second most common bone malignancy in adults, and is characterized by highly contrasting prognoses, due to the heterogeneity of its tumor subtypes. Currently, histological evaluation is the only means of grading chondrosarcoma, assessing prognosis and choosing proper treatments after tumor resection. Yet, the histopathological assessment of chondrosarcoma grade is not considerably reliable for diagnosis and prognosis, and methods for a quantitative and objective evaluation of tumors are urgently needed. For instance, biomarkers to classify and grade chondrosarcoma, which may improve the reliability of its diagnosis and predictions of its clinical behavior for therapeutic management, are widely sought [1,2,4,41]. Proteomic approaches have contributed to the characterization of protein profiling for various cancer types [42,43,44]. Quantitative proteomics provides large-scale differential protein abundance in healthy and tumorous samples, making this technique particularly well-suited to the investigation of molecular mechanisms and the discovery of potential predictive and prognostic biomarker candidates. Nevertheless, the high complexity of serum and other biological fluids renders it difficult to achieve the proteomic identification of differentially secreted proteins, including potential biomarkers. High-abundance proteins such as albumin and immunoglobulins mask the presence of these proteins. The depletion of albumins, immunoglobulins and other serum proteins from plasma has been reported to reduce sample complexity and allow mass-spectrometry detection of low-abundant proteins that would not be detected in a plain proteome analysis, as their signal would be covered by that of high-abundant serum proteins [45,46,47]. To date, there are few data on chondrosarcoma malignancies and no potential biomarkers have been identified through proteomic analysis [41,48]. Given these limitations, we used an in vitro approach to identify by proteomics proteins that may be differentially secreted during chondrosarcoma development.

A common feature shared by solid tumors, including chondrosarcoma, is that tumor malignancy occurs in response to a hypoxic tumor microenvironment, leading to cancer survival, progression and metastatic process [6,9]. Hypoxia-inducible factors (HIFs) orchestrate the hypoxia signalling pathway inducing the expression of hypoxia-responsive genes involved in angiogenesis and metabolism, such as glycolytic enzymes and glycogenic enzymes [7,8,49]. HIF-1α correlates with disease progression and prognosis. Thus, we treated SW 1353 chondrosarcoma cells with DMOG, a compound that stabilizes HIF-1α, thereby mimicking hypoxia [14,50,51], and performed a quantitative secretome analysis to identify proteins that were differentially released in response to hypoxia. SW 1353 cells represent a model that we have extensively investigated and the secretome of which we have fully characterized [52,53]. We found that approximately 50% of the total proteins were altered in response to hypoxia. Our method, which is particularly well-suited to identifying proteins secreted by unconventional pathways in addition to canonical secreted proteins, identified a large number of cytosolic and exosome-associated proteins in the conditioned media of DMOG-treated chondrosarcoma cells [15]. The majority of them were involved in the organization of the ECM (Table 5), in line with the established role of hypoxia in ECM remodelling and cancer progression [54,55]. A large number of hypoxia-regulated proteins that we identified in the secretome of chondrosarcoma cells were associated with EV compartment. Growing evidence has shown that hypoxia is also involved in EV biogenesis and release in different cell types, including cancer cells [56]. EVs are lipid-bound vesicles secreted by cells into the extracellular space [29]. In the tumor microenvironment, EVs are a means of paracrine signalling since the earlier stages, promoting progression, invasiveness and metastasis development [57,58]. It is now well-established that hypoxia affects the number, size, and molecular cargo of EVs in cancer [59]. Despite differences in cell models and hypoxic treatments, the number of EVs increased upon hypoxic condition in breast cancer, prostate cancer and ovarian cancer [60,61,62,63]. This is consistent with our results in chondrosarcoma, in which exosome release upon chemical hypoxia treatment increased more than twofold.

Among the proteins altered by DMOG treatment, we validated GAPDH by western blotting as an orthogonal method. GAPDH is a glycolytic enzyme that catalyzes the conversion of glyceraldehyde 3-phosphate to 1,3-diphosphoglycerate, but it is considered to be a moonlighting protein because of its multifunctionality and different subcellular localization, not only ascribed to cytoplasm, but also to the nucleus, membrane, extracellular region and EVs [64,65]. Indeed, its role in EV biogenesis has recently emerged [38]. In various cancer types, GAPDH expression and protein levels are increased, resulting in the alteration of functional roles and improving progression and invasiveness [31,33,66]. In agreement, we found increased levels of GAPDH in the extracellular milieu. Since it has been reported that GAPDH is secreted through an unconventional mechanism and is also strongly associated with EVs (Appendix A), we further investigated its extracellular compartmentalization [35,37,64] to confirm that GAPDH could also be secreted by this dual mechanism in chondrosarcoma cells. GAPDH and other proteins whose secretion was augmented in response to hypoxia, including CNBP and SNRPA, could represent proteomic signatures of chondrosarcoma and could be developed as chondrosarcoma biomarkers. More information about peptides identified in this analysis can be found in Appendix A.

In conclusion, our study analyzed the effects of hypoxia on the secretome of chondrosarcoma, a tumor for which reliable biomarkers and effective therapies other than surgical resection are not currently available. In addition to molecular processes that are altered in response to hypoxia and which provide new information about chondrosarcoma progression, we identified a number of proteins, including GAPDH, that may be further developed as predictive and/or prognostic biomarkers for chondrosarcoma.

## Figures and Tables

**Figure 1 proteomes-10-00025-f001:**
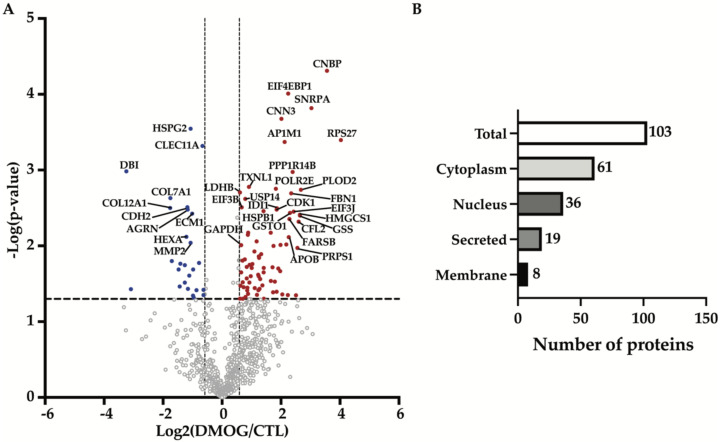
Analysis of secretome composition of DMOG-treated SW 1353 cells. (**A**) Volcano plot showing the −log10 of *p*-values versus the log2 of protein ratio between DMOG-treated (DMOG) and control SW 1353 cells (CTL) of 856 proteins (*n* = 3). Proteins significantly regulated are displayed as the filled dots above *t*-test based *p*-value < 0.05 (black dashed horizontal line) and fold change DMOG/CTL higher or lower than 50% (black dashed vertical lines). Red dots correspond to more abundant proteins, blue dots to less abundant proteins in the secretome of DMOG-treated cells. (**B**) Subcellular location of altered proteins detected in the secretome of DMOG-treated SW 1353 cells according to Uniprot annotation.

**Figure 2 proteomes-10-00025-f002:**
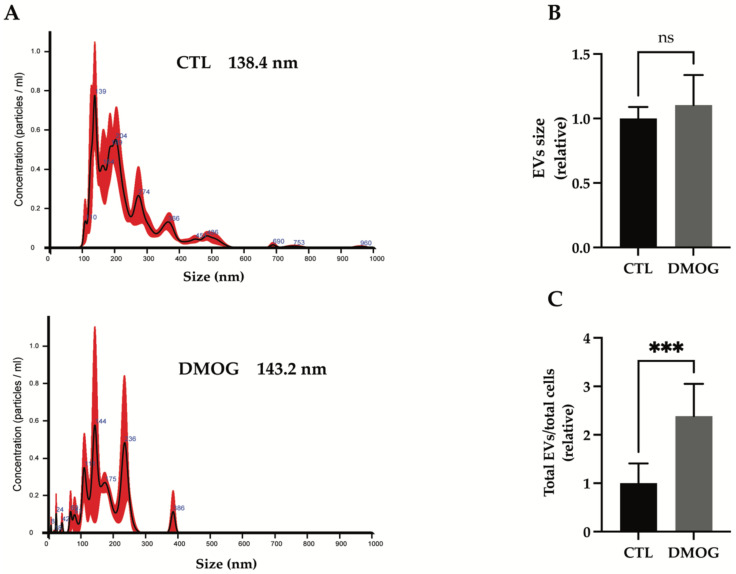
Characterization of isolated EVs from conditioned media of SW 1353 DMOG-treated or control cells. (**A**) Representative profiles of size distribution of EVs determined by nanoparticle tracking analysis (NTA), mode of the size distributions is indicated in the figure. Bar graphs show calculation of the mode of the size distribution (**B**) and total number of EVs per total number of cells (**C**) from SW 1353 control and DMOG-treated; analyses are displayed as mean values ± standard deviation (*** *p* < 0.001, Student’s *t*-test; *n* = 7).

**Figure 3 proteomes-10-00025-f003:**
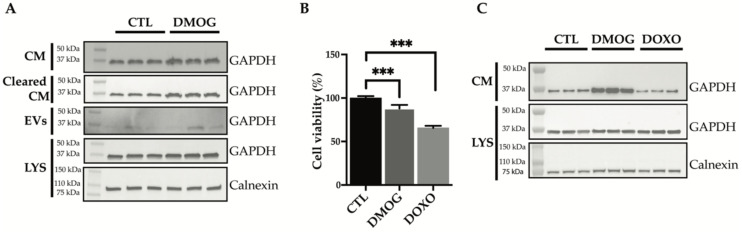
Analysis of GAPDH levels by immunoblotting. (**A**) Immunoblots showing GAPDH protein abundance in the conditioned media (CM), in the conditioned media cleared of extracellular vesicles (Cleared CM), in the extracellular vesicles (EVs) and in the cell lysate (LYS) of SW 1353 cells treated with or without DMOG. Calnexin was used as a loading control. (**B**) Cell viability analysis of SW 1353 cells treated with DMOG or doxorubicin (DOXO). DMSO-treated cells were used as controls (CTL). Data are represented as % of the mean vs. control (*n* = 6). (**C**) Immunoblots showing GAPDH protein abundance in the conditioned media (CM) and lysate (LYS) of SW 1353 cells treated with DMSO (CTL), DMOG or doxorubicin (DOXO). Calnexin was used as a loading control (*p* < 0.001).

**Table 1 proteomes-10-00025-t001:** List of selected proteins significantly increased in the secretome of SW 1353 cells in response to DMOG treatment.

Protein Name	Protein ID	Gene Name	*p*-Value	Fold Change	Ratio
Cellular nucleic acid-binding protein	P62633	CNBP	2.04 × 10^−4^	11.783	3.559
Eukaryotic translation initiation factor 4E-binding protein 1	Q13541	EIF4EBP1	1.02 × 10^−4^	4.738	2.244
U1 small nuclear ribonucleoprotein A	P09012	SNRPA	6.55 × 10^−3^	8.177	3.032
Calponin-3	Q15417	CNN3	4.74 × 10^−3^	4.041	2.015
40S ribosomal protein S27	P42677	RPS27	2.48 × 10^−3^	16.358	4.032
AP-1 complex subunit mu-1	Q9BXS5	AP1M1	2.35 × 10^−3^	4.356	2.123
Protein phosphatase 1 regulatory subunit 14B	Q96C90	PPP1R14B	9.45 × 10^−2^	5.259	2.395
Thioredoxin-like protein 1	O43396	TXNL1	5.99 × 10^−2^	1.883	0.913
DNA-directed RNA polymerases I, II, and III subunit RPABC1	P19388	POLR2E	5.66 × 10^−2^	3.545	1.826
Procollagen-lysine,2-oxoglutarate 5-dioxygenase 2	O00469	PLOD2	5.50 × 10^−2^	6.383	2.674
L-lactate dehydrogenase B chain	P07195	LDHB	5.07 × 10^−2^	1.525	0.609
Fibrillin-1	P35555	FBN1	4.93 × 10^−2^	5.079	2.345
Ubiquitin carboxyl-terminal hydrolase 14	P54578	USP14	4.16 × 10^−2^	1.73	0.790
Eukaryotic translation initiation factor 3 subunit B	P55884	EIF3B	3.22 × 10^−2^	1.590	0.669
Isopentenyl-diphosphate Delta-isomerase 1	Q13907	IDI1	3.13 × 10^−2^	3.634	1.862
Cyclin-dependent kinase 1	P06493	CDK1	3.01 × 10^−2^	3.611	1.852
Heat shock protein b-1	P04792	HSPB1	2.87 × 10^−2^	2.650	1.406
Eukaryotic translation initiation factor 3 subunit J	O75822	EIF3J	2.83 × 10^−2^	5.377	2.427
Glutathione S-transferase omega-1	P78417	GSTO1	2.72 × 10^−2^	4.930	2.302
Hydroxymethylglutaryl-CoA synthase, cytoplasmic	Q01581	HMGCS1	2.57 × 10^−2^	6.231	2.639
Glutathione synthetase	P48637	GSS	2.51 × 10^−2^	6.250	2.644
Phenylalanine--tRNA ligase beta subunit	Q9NSD9	FARSB	2.26 × 10^−2^	4.877	2.286
Cofilin-2	Q9Y281	CFL2	2.08 × 10^−2^	6.046	2.596
Leucine-rich repeat flightless-interacting protein 1	Q32MZ4	LRRFIP1	1.49 × 10^−2^	1.847	0.886
X-ray repair cross-complementing protein 5	P13010	XRCC5	1.49 × 10^−2^	3.137	1.649
UMP-CMP kinase	P30085	CMPK1	1.40 × 10^−2^	1.833	0.875
Apolipoprotein B-100	P04114	APOB	1.30 × 10^−2^	4.802	2.264
Eukaryotic translation initiation factor 4 gamma 1	Q04637	EIF4G1	1.15 × 10^−2^	2.2550	1.173
Actin-related protein 2/3 complex subunit 3	O15145	ARPC3	1.04 × 10^−2^	4.534	2.181
60S ribosomal protein L8	P62917	RPL8	1.03 × 10^−2^	3.955	1.984
Glyceraldehyde-3-phosphate dehydrogenase	P04406	GAPDH	1.02 × 10^−2^	1.565	0.646

Protein name: proteins that increased in the secretome of SW 1353 cells upon DMOG treatment with a *t*-test *p*-value lower than 0.01 and a fold change higher than 50%. Protein ID: UniProt accession number of the protein. Gene name: Uniprot gene name associated with each protein. *p*-value: for three biological replicates. Fold change: LFQ ratio between the mean of LFQ values of DMOG-treated SW 1353 and controls (*n* = 3). Ratio: mean ratio of label-free quantification intensities between DMOG-treated and control SW 1353 cells (*n* = 3).

**Table 2 proteomes-10-00025-t002:** List of selected proteins significantly decreased in the secretome of SW 1353 cells in response to DMOG treatment.

Protein Name	Protein ID	Gene Name	*p*-Value	Fold Change	Ratio
Basement membrane-specific heparan sulfate proteoglycan core protein 2	P98160	HSPG2	3.51 × 10^−3^	0.477	1.069
C-type lectin domain family 11 member A	Q9Y240	CLEC11A	2.08 × 10^−3^	0.632	0.663
Collagen alpha-1(VII) chain	Q02388	COL7A1	4.26 × 10^−2^	0.297	1.753
Cadherin-2	P19022	CDH2	3.23 × 10^−2^	0.442	1.178
Collagen alpha-1(XII) chain	Q99715	COL12A1	3.16 × 10^−2^	0.295	1.760
Agrin	O00468	AGRN	3.01 × 10^−2^	0.446	1.166
Extracellular matrix protein 1	Q16610	ECM1	2.65 × 10^−2^	0.495	1.013
Beta-hexosaminidase subunit alpha	P06865	HEXA	1.31 × 10^−2^	0.433	1.209
72 kDa type IV collagenase	P08253	MMP2	1.10 × 10^−2^	0.478	1.066

Protein name: proteins that decreased upon DMOG treatment with a *t*-test *p*-value lower than 0.01 and a fold change higher than 50%. Protein ID: UniProt accession number of the protein. Gene name: Uniprot gene name associated with each protein. *p*-value: for three biological replicates. Fold change: LFQ ratio between the mean of LFQ values of DMOG-treated SW 1353 and controls (*n* = 3). Ratio: mean ratio of label-free quantification intensities between DMOG-treated and control SW 1353 cells (*n* = 3).

**Table 3 proteomes-10-00025-t003:** List of the most enriched terms after TRANSFAC library analysis.

TRANSFAC		
Term	*p*-Value	Proteins
HIF1A (human)	1.50 × 10^−4^	EIF5A, PRPS1, SNRPN, DBNL, RPS6, CSTF2, TXNL1, PLOD2, ACTN4, PPM1G, LDHB, TUBA1C, PSMC3, OAF, POLR2E, MAPK1, MCM6, EIF3A, EIF4G1, AP1M1
GATA1 (human)	3.76 × 10^−4^	APP, IDI1, PRPS1, ECM1, CSTF2, PPM1G, SCRN1, CFL2, CAPN2, CCT8, PDLIM5, EIF5A, NUDC, SNRPN, DBNL, HMGCS1, XRCC5, NONO, MMP2, RPS6, TXNL1, CLEC11A, HSPG2, ACTA1, CDK1, CMPK1, EIF4G1

Table displaying term names, *p*-values and proteins belonging to the term of the JASPAR PWMs enrichment analysis (terms with a *p*-value below 1.0 × 10^−3^ are listed).

**Table 4 proteomes-10-00025-t004:** List of the most enriched terms in the GO Biological Process analysis.

GO Biological Process 2021		
Term	*p*-Value	Proteins
Extracellular matrix organization (GO:0030198)	4.81 × 10^−9^	APP, MMP2, COL12A1, TNC, FN1, PLOD2, NID1, HSPG2, COL7A1, CAPN2, AGRN, PRSS2, FBN1
Cellular protein metabolic process (GO:0044267)	3.01 × 10^−8^	APP, MMP2, RPS6, TNC, FN1, PLOD2, PLAT, RPL8, RPS27, CDH2, APOB, EIF4G1, FARSB, FBN1

Table displaying term names, *p*-values and proteins associated with terms in the GO Biological Process analysis (terms with a *p*-value below 1.0 × 10^−7^ are listed).

**Table 5 proteomes-10-00025-t005:** List of the most enriched terms in the GO Cellular Component analysis.

GO Cellular Component 2021		
Term	*p*-Value	Proteins
Secretory granule lumen (GO:0034774)	8.95 × 10^−9^	LGALS3BP, APP, ECM1, DBNL, XRCC5, TUBB, FN1, ACTN4, PSMC3, IMPDH2, MAPK1, CCT8, PRSS2
Intracellular organelle lumen (GO:0070013)	5.79 × 10^−8^	APP, DBNL, OAT, GSR, COL12A1, TNC, FN1, DBI, HSPG2, PSMC3, CDH2, COL7A1, IMPDH2, CDK1, MAPK1, CCT8, AGRN, APOB, FBN1
Collagen-containing extracellular matrix (GO:0062023)	7.84 × 10^−8^	LGALS3BP, ECM1, MMP2, COL12A1, TNC, FN1, PLAT, NID1, HSPG2, CDH2, COL7A1, AGRN, FBN1
Focal adhesion (GO:0005925)	9.68 × 10^−8^	TNC, HSPB1, ACTN4, RPL8, HSPG2, CNN3, PROCR, CDH2, ARPC3, CAPN2, ITGBL1, MAPK1, TLN1

Table displaying term names, *p*-values and proteins associated with terms in the GO Cellular Component analysis (terms with a *p*-value below 1.0 × 10^−7^ are listed).

**Table 6 proteomes-10-00025-t006:** List of the most enriched terms after the Jensen COMPARTMENT library analysis.

Jensen COMPARTMENT		
Term	*p*-Value	Proteins
Extracellular exosome	1.91 × 10^−22^	LGALS3BP, APP, EIF4A1, ECM1, COL12A1, HEXA, HSPB1, PLOD2, PLAT, DBI, TUBB6, TUBA1A, CDH2, CFL2, CAPN2, EIF5A, DBNL, GSTO1, TUBB, EPDR1, ACTN4, HSPG2, SND1, DDB1, PROCR, ACTA1, MTHFD1, OAF, CMPK1, TLN1, GART, GAPDH, USP14, RAB1B, NID1, LDHB, PCBP1, MAPK1, CCT8, APOB, AP1M1, HSPA4, GSS, GSR, TXNL1, FN1, GGCT, ARPC3, IMPDH2, CDK1, ACO1, AGRN, RBMX, FBN1, AARS, EIF3B
Extracellular vesicle	2.45 × 10^−22^	LGALS3BP, APP, EIF4A1, ECM1, COL12A1, HEXA, HSPB1, PLOD2, PLAT, DBI, TUBB6, TUBA1A, CDH2, CFL2, CAPN2, EIF5A, DBNL, GSTO1, TUBB, EPDR1, ACTN4, HSPG2, SND1, DDB1, PROCR, ACTA1, MTHFD1, OAF, CMPK1, TLN1, GART, GAPDH, USP14, RAB1B, NID1, LDHB, PCBP1, MAPK1, CCT8, APOB, AP1M1, HSPA4, GSS, GSR, TXNL1, FN1, GGCT, ARPC3, IMPDH2, CDK1, ACO1, AGRN, RBMX, FBN1, AARS, EIF3B
Extracellular organelle	2.49 × 10^−22^	LGALS3BP, APP, EIF4A1, ECM1, COL12A1, HEXA, HSPB1, PLOD2, PLAT, DBI, TUBB6, TUBA1A, CDH2, CFL2, CAPN2, EIF5A, DBNL, GSTO1, TUBB, EPDR1, ACTN4, HSPG2, SND1, DDB1, PROCR, ACTA1, MTHFD1, OAF, CMPK1, TLN1, GART, GAPDH, USP14, RAB1B, NID1, LDHB, PCBP1, MAPK1, CCT8, APOB, AP1M1, HSPA4, GSS, GSR, TXNL1, FN1, GGCT, ARPC3, IMPDH2, CDK1, ACO1, AGRN, RBMX, FBN1, AARS, EIF3B

Table displaying term names, *p*-values (significant *p*-value < 0.05) and proteins associated with terms in the Jensen COMPATMENT library (terms with a *p*-value below 1.0 × 10^−20^ are listed).

## Data Availability

The data supporting results in this study are included within the article and available in Appendix A.

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
