# Peer review of "High-Resolution Secretome Analysis of Chemical Hypoxia Treated Cells Identifies Putative Biomarkers of Chondrosarcoma"

_proteomes, 2022, doi:10.3390/proteomes10030025_

Round 1
Reviewer 1 Report
I appreciate the attention of the authors for the suggested revisions.
Author Response
We would like to thank the reviewer for his/her positive comments and appreciation of our manuscript.
Reviewer 2 Report
The manuscript entitled High-resolution secretome analysis of chemical hypoxia treated cells identifies putative biomarkers of chondrosarcoma after revision made by the authors, increased its value, however it still need to be improved:
1 1. In the introduction (line 88-95), the authors present in fact a summary of the study. The introduction should not describe results or conclusions. However, it should contain the hypothesis and the aim of the study, which is missing here.
2. In response to my comment on the cell line used in the study, the authors wrote that "SW1353 cells represent a model that we have extensively investigated and the secretome of which we have fully characterized". This is very valuable information (as well as the cited articles). It would be good if they were also included in the text of the manuscript.
3. The authors did not refer to my comment on the sentence: "the text describes the method, not the results" (line 196-198 in last version of manuscript)
4.line 253-256 - these sentences should not be in the results section, but in the discussion
5. The authors used Calnexin as a marker for WB, which is obviously good for the lysate, but (as emphasized by the authors) not for EV and medium. In a situation where we cannot apply the marker to all membranes, it is worth using 2,2,2-Trichloroethanol to improof that the applied samples have the same concentration.
Author Response
- In the introduction (line 88-95), the authors present in fact a summary of the study. The introduction should not describe results or conclusions. However, it should contain the hypothesis and the aim of the study, which is missing here.
While respecting the reviewer’s comment, I disagree with it as briefly mentioning results and conclusions in the very last paragraph of the introduction is a very common practice in research article writing (for instance [1; 2]). This helps in engaging the attention of the reader and encourages the reader to read further.
- In response to my comment on the cell line used in the study, the authors wrote that "SW1353 cells represent a model that we have extensively investigated and the secretome of which we have fully characterized". This is very valuable information (as well as the cited articles). It would be good if they were also included in the text of the manuscript.
This sentence, as well as citations, have been now added (line 361-363)
- The authors did not refer to my comment on the sentence: "the text describes the method, not the results" (line 196-198 in last version of manuscript)
We apologize with the reviewer for the inconvenience, we must have missed it somehow. A comprehensive description of the proteomic workflow is indeed detailed in the Materials and Methods section. Nevertheless, we briefly reiterated the experimental set up at the beginning of the Results section in order to facilitate the readers’ comprehension of following results.
4.line 253-256 - these sentences should not be in the results section, but in the discussion
We respect the reviewer’s opinion, but we believe that a conclusive sentence to summarize the results, or part of the results, can help the reader’s comprehension of the study, and it is commonly used in article writing [1; 2]
- The authors used Calnexin as a marker for WB, which is obviously good for the lysate, but (as emphasized by the authors) not for EV and medium. In a situation where we cannot apply the marker to all membranes, it is worth using 2,2,2-Trichloroethanol to improof that the applied samples have the same concentration.
We would like to thank the reviewer for his/her comments, and we will surely try this method for our future characterizations (actually, we would appreciate more details about the method). Indeed, proper loading controls for proteins in the secretome are currently missing. Despite big effort, we have not been able to validate any of such control secreted proteins (i.e. progranulin, clusterin, etc) as any kind of treatment seems to alter protein secretion. Given that our validation experiments followed a proteomic analysis in which the total amount of proteins in the secretome was normalized by the mass spectrometer, we believe that normalizing secreted proteins on levels of cell lysate proteins expressed by housekeeping genes, such as calnexin, actin, would be accurate enough to support our conclusions.
References
- Grieve AG, Xu H, Künzel U, Bambrough P, Sieber B, Freeman M. Phosphorylation of iRhom2 at the plasma membrane controls mammalian TACE-dependent inflammatory and growth factor signalling. Elife. 2017 Apr 22;6:e23968. doi: 10.7554/eLife.23968. PMID: 28432785; PMCID: PMC5436907.
- Colombo A, Hsia HE, Wang M, Kuhn PH, Brill MS, Canevazzi P, Feederle R, Taveggia C, Misgeld T, Lichtenthaler SF. Non-cell-autonomous function of DR6 in Schwann cell proliferation. EMBO J. 2018 Apr 3;37(7):e97390. doi: 10.15252/embj.201797390. Epub 2018 Feb 19. PMID: 29459438; PMCID: PMC5881626.
This manuscript is a resubmission of an earlier submission. The following is a list of the peer review reports and author responses from that submission.
Round 1
Reviewer 1 Report
In this manuscript the authors studied the secretome of chondrosarcoma cell line in response to a chemically induced hypoxia. The authors use label free proteomics to find differentially secreted proteins in response to hypoxia. Results from the differential expression analysis indicated that the chondrosarcoma cell line secreted more of exosome-associated proteins. The authors also further show that EVs secreted might be exosomes based on nanoparticle tracking analysis. Finally the authors using westren blot validate the differential secretion of GAPDH in both secretome and EV fraction as well. The manuscript is clearly written and easy to understand.
Following are my comments.
1) The authors used DMOG (500uM) to induce chemical hypoxia. I wonder about the status of the cells after treating with this drug. For example the shape and viability of the cells. 500uM seems to be a high concentration and if the cells are in a bad shape following the treatment, it can drastically affect the secretome. The authors should provide more information about this aspect.
2) Also considering the above fact, that authors should have also used another irrelavent cell line in the study as a treated control. This would establish the background secretome in response to hypoxia. With the current data, it is hard to correlate the conclusion to chondrosarcoma.
3) Using quantitative proteomics the authors show that the exosome-associated proteins are differentially secreted and further validate GAPDH in both secretome and EVs. I wonder why the authors chose to use GAPDH, while there are many other proteins (Table1) that were differentially secreted to a significant extent. Availability of antibodies is not a good explanation, as one could use targeted proteomics for example.
4) It would have been very interesting, if the authors would have also preformed quantitative proteomics of the EVs isolated from the secretome. This data could have added an additional layer of validation of the conclusions.
Inconclusion, this is a very well written paper and few more justifications would have been a great addition to the conclusions.
Reviewer 2 Report
The manuscript represents an interesting dataset about a rare tumor type.
It is clearly and well written. Methods are well described and figures and tables are clear.
I suggest the authors to address the following points within their manuscript:
- Peptide sequences should be reported, e.g. in a supplementary table. I strongly encourage the authors to upload their results to public repository databases like proteomeXchange to contribute with their work to the efforts of the community.
- How many biological (cell cultures) and technical (injections) has been used?
- How was normalization performed across experiments (quantitative data from treated vs non treated cell lines)? What was the cell number/confluence per group at the timepoint when the secretome has been collected?
- How much did the DMOG treatment affect the cell survival = toxicity? How did the author distinguish “true” hypoxia effects from cell deaths/DMOG induced toxicity?
- What was the injection volume on the MS per samples? How was liquid chromatography performed (e.g. flow rate, buffers, column composition, etc.)
- Line 49, page 2: the information about death rate for the metastatic state of the disease is not in accord with the common mortality statistic of the cancer. If these data are really needed, please correct the numbers in accord to the correct epidemiologic data and report the reference.
- The authors report that the motivation of the work was about prognostic biomarkers and possibly treatment targets. Please discuss how their results do fit to their original motivation and what will be the next steps to achieve their goal.
Reviewer 3 Report
Identification of biomarkers are important for clinical diagnosis of disease. Various molecular markers are identified for different types of cancers. However, for chondrosarcoma, accurate prediction is not possible. Accurate prediction of markers is needed for deciding the better treatment procedure. This paper use mainly proteomics analysis to identify markers. However, there are some concerns.
1) In table 1, include fold change.
2) Figure 3, description is not sufficient.
3) Major concern is for using GADPH as biomarkers because this is used as an inter control for various tissue assays. Also, ratio for this is low according to table 1. What is the rational behind using GADPH as a marker. There are other protein for which ratio is very high. Alpha-methylacyl-CoA racemase periostin and VEGF isoforms are identified as a biomarkers for chondrosarcoma.
4. Validation of the proposed marker is missing.
5. Proteins associated with HIF should be considered for identifying biomarkers. Discussion about this is also missing.
Reviewer 4 Report
In the manuscript titled High-resolution secretome analysis of chemical hypoxia treated cells identifies putative biomarkers of chondrosarcoma, Spanò DP. et al. analyzed effects of chemically induced hypoxia on the secretome of chondrosarcoma cell line SW 1353. The authors also suggested the possibility of using the GAPDH protein as a potential biomarker in the diagnosis and treatment of chondrosarcoma.
The analyzes presented by Spanò DP. et al. are of great scientific importance, due to the difficulties in the treatment and proper determination of the chondrosarcoma grade.
The manuscript presented for review is written very concisely and logically, however requires corrections:
The main disadvantage of the study is the use of only one chondrosarcoma cell line. It would be good to confirm these results using e.g. JJ012 or ch-2879 lines, or at least explain why the research was carried out on the SW1353 cell line. For example, have there been any pilot studies on the basis of which this particular cell line was selected?
Another concern is Western Blot control. Was Calnexin also tested in conditioned medium and EVs?
It is also necessary to add a description of the statistics as a separate subsection. Partly such a description is given in point 2.3, but more information is needed, e.g. what statistical program was used for the analysis.
There is also no information about the number of biological repetitions. For example, on how many repetitions the secretome tests were performed
Minor points:
- a) The abstract should include the name of the tested cell line
- b) Material and Methods:
* 2.1:
- „SW 1353 cells” the name „chondrosarcoma” is missing;
- „DMSO was used as a control” – the sentence needs to be corrected, it does not explain that the DMOG has been replaced by DMSO.
*2.5:
- does the „absence of DMOG: mean DMSO? (line 153);
- „separate from EVs” (line 154) - what method was used?
- what transfer was made, which secondary antibodies were used
- the text of 2.5 subsection is very confusing and requires editing
- c) Results:
- line 169-171: the text describes the method, not the results
- figure 1B: x-axis description missing, accession date information
- figure 2: lack of „a” in the figure
- figure 3: lack of „LYS” explanation in the description of the figure